# Landscape Modifications Ascribed to El Niño Events in Late Pre-Hispanic Coastal Peru

Marco Delle Rose

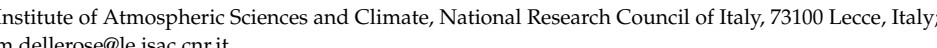

Institute of Atmospheric Sciences and Climate, National Research Council of Italy, 73100 Lecce, Italy; m.dellerose@le.isac.cnr.it

**Abstract:** Coastal Peru, one of the driest deserts in the world, is a key region to investigate the connection between climate processes and Earth surface responses. However, the trends in space and time of the landscape effects of El Niño events throughout the last millennium are hard to outline. A deeper understanding of geological and archaeological data in pre-Hispanic time can help to shed light on some critical questions regarding the relationship between such a coupled atmosphere–ocean phenomenon and landscape modifications. The bibliographic sources required for this purpose are scattered throughout various disciplines, ranging from physical to human sciences, and thus comprehensive databases were used to identify and screen relevant studies. The performed examination of these documents allowed us to assess strengths and weaknesses of literature hypotheses and motivate additional studies on targeted research objectives.

**Keywords:** desert landscape; coastal plain; paleoflood record; El Niño proxies; debris flow; slack-water deposit; braided streams; desert pavement; regolith denudation

## 1. Introduction

Due to the dynamic of the El Niño-Southern Oscillation (ENSO), the coastal desert of Peru is a key region to investigate the connections between climate processes and Earth surface responses [1–3]. El Niño precipitation events cause abrupt and rapid landscape modifications on the whole central Pacific coast of South America [4,5]. However, the trends in space and time of the landscape effects of El Niño events on the coastal Peru throughout the last millennia are hard to outline [6,7].

Coastal Peru has recently experienced dramatic landscape changes and asset destruction during precipitation events due to El Niño. Widespread landslides and extensive floods are the more relevant hydrogeomorphic signatures [8–12]. With reference to the late Holocene, geoarchaeological studies are crucial to explore El Niño proxies and paleoflood record [4,9,13]. As a matter of fact, several archaeological and geological studies have focused on the strong impact on settlements and irrigation systems of severe precipitation events named "Super" and "Mega" El Niños [14–19], sometimes believed to be the cause of the collapse of centuries-old cultures [20–23]. However, for some case studies, alternative interpretations involving gradual evolution of the landscape or questioning the occurrence itself of the catastrophic events have been provided in the literature [24–26].

A deeper understanding of the geological and archaeological data can help to shed light on some critical questions about the relationship between landscape modifications and El Niño events. This review aims to explore the period from the rise of "modern" periodicity of ENSO [13,27] and beginning of the Little Ice Age (roughly coincident with the arrival of the Spanish conquistadors) [28,29], herein called "late pre-Hispanic". The bibliographic sources required for this purpose are scattered throughout the literature of various disciplines ranging from physical to human sciences. Thus, the identification and screening of pertinent documents (Sections 3 and 4) have been performed by the citation databases of Web of Science (WoS) and Scopus, in accordance with the Preferred Reporting Items for

Systematic Reviews and Meta-Analyses statement (PRISMA) [30,31]. Episodes of landscape change ascribed to severe precipitations in the late pre-Hispanic time are discussed, and their consistencies and inconsistencies are exposed (Section 5). Some examples of variation in landscape response due to extensive human intervention are also reported. The careful examination of geoarchaeological data allowed strengths and weaknesses of literature hypotheses on landscape effects of El Niño events to be assessed and motivates additional studies on targeted research objectives (Sections 5 and 6).

## 2. Background Knowledge about the Topic

Due to the interdisciplinary nature of the subject matter, some basic concepts must be given before addressing the Materials and Methods (Section 3).

### 2.1. Physical Setting

The landscape of coastal Peru (Figure 1) is characterized by hill ridges—the western Andean offshoots named *lomas*—made up of pre-Tertiary carbonate, igneous, and metamorphic rocks, and flat areas crossed by rivers—the *pampas*—made up by Tertiary-Quaternary clastic rocks produced from the dismantling of the Andes during orogenesis. Both are devoid of soil and vegetation and partially covered by deposits produced by eolian processes (dunes, desert pavements, and reg soils). Typical landforms are the *quebradas*, which dry braided stream and ravine systems mainly located at the Andean footzone (over alluvial fans) or next to the coast.

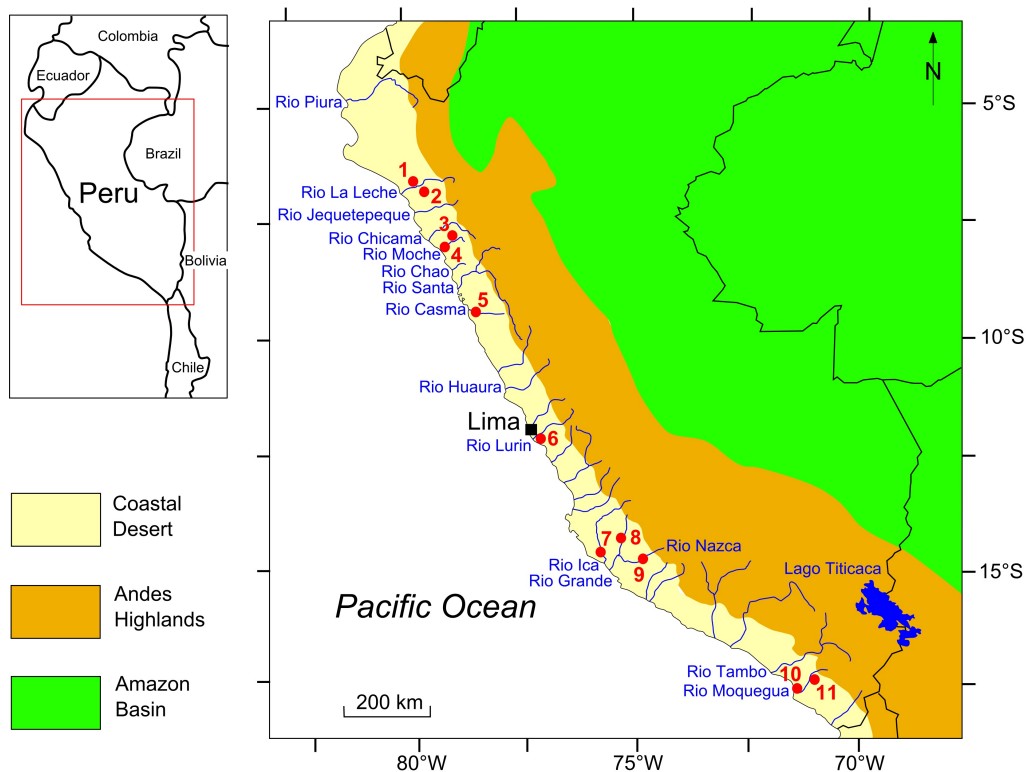

**Figure 1.** Main geographical regions of Peru. Numbers indicate the location of the cases discussed in the text; 1, Batan Grande; 2, Racarumi; 3, Caballo Muerto and Quebrada de los Chinos; 4, Huacas de Moche; 5, Casma and Cerro Sechin; 6, Pachacamac and Urpi Kocha; 7, Samaca; 8, Palpa; 9, Cahuachi; 10, Quebrada Miraflores; 11, Rio Muerto.

Coastal Peru (3°30′–18°30′ S, 81°30′–70°30′ W) is one of the driest regions in the world. Such a desert environment results from the SE Pacific anticyclone and the Humboldt Current coming from the Antarctic Sea, because they both prevent rain [32,33]. Annual precipitation values are rather uniform along the entire coast, averaging 10–25 mm per year.

The desert along the Pacific is narrow, only about 120 km wide before the land rises into the Andes Highlands, where precipitation increases with elevation. Tens of perennial or seasonal rivers, coming down from the Andes, pass through the desert on the way to the ocean (Figure 1). They allow both biological life and human settlement within and in the surrounding riparian oases. Once every 7–15 years, an El Niño brings warm sea surface temperatures and torrential rainfall on the coastal region, breaking the hyperarid state. Within the last century, severe El Niño events occurred during the years 1925/1926, 1940/1941, 1957/1958, 1972/1973, 1982/1983, 1997/1998, and 2015/2016, the latter with an extraordinary prolongation in 2017 [7,10,34–37].

*2.2. El Niño Event and Landscape Response*

The term El Niño was initially applied to a weak warm ocean current that runs southward along the Ecuador and Peru coastline around Christmastime. It is the warm phase of the El Niño–Southern Oscillation (ENSO), an interannual fluctuation in sea surface temperature and air pressure across the equatorial Pacific Ocean [38–41]. The cold phase is called La Niña, and the neutral phase is the intermediate one. Such coupled atmosphere–ocean phenomena dominate the interannual variability of planetary climate system and cause strong drought and flood around the world [42,43]. ENSO has dramatically changed in occurrence and magnitude during the Holocene, increasing in frequency to reach its current features about 3.0 Ka ago [27,44,45], roughly corresponding with the beginning of the Early Horizon (ca. 800–200 BCE).

Coastal Peru is responsible for a direct correspondence among El Niño severity, flooding, and landscape modifications. During El Niño, the South Pacific anticyclone weakens while the northern boundary of the Humboldt current migrates southward. In such a condition, significant precipitation is able to reach the coastal desert, where it is reinforced due to the rain-shadow effect of the Andes [17]. However, the flood magnitude does not affect the different desert areas equally [34,35,37]. Latitude 12° S roughly divides north-central coastal Peru, which is hardest hit by El Niño events, from southern coastal Peru, which is usually less affected [46]. Nevertheless, large events, such as that in 1925/1926, can extend much farther south [47].

Severe precipitation and flooding may be also produced by other atmospheric patterns. The eastern desert margin is involved in monsoonal fluctuations, and thus thunderstorms can occasionally reach it and cause convective precipitation, especially over southern Peru [48,49]. Again, enhanced precipitation across the Andes highlands during the La Niña event may produce flooding on main valleys [50,51]. From the perspective of paleoflood analysis, such processes make it more challenging to detect the triggering cause of hydrogeomorphic events preserved in geoarchaeological sequences [4,9].

The strength of ENSO events is measured by conventional physical indices. Regional Sea Surface Temperature (SST) indices and the surface atmospheric pressure-based Southern Oscillation index (SOI) are the most widely used indexes to classify ENSO events. However, there is no agreement among scholars about which index best defines ENSO strength, timing, and duration, and fits severe precipitations satisfactorily [38,42,52–54]. To address the change in frequency and intensity of events ascribed to El Niño events that hit the central Pacific coast of South America in pre-instrumental time, various approaches have been used, including geoarchaeological studies [10,55]. After the event of 1982/1983, scholars have focused on the power and occurrence of extremely strong events. Events with a recurrence interval up to 400 years were labeled as Super El Niño, while the ones that happen once every 1000 years as Mega El Niño [19,56].

The frequency and spatial variability of El Niño events is very difficult to establish and predict. Historical climatic and hydrological records may reflect specific atmospheric conditions of each valley and parts of them. To give some examples, the Lower Chicama Valley (northern Peru, Figure 1) saw extensive flooding as a result of the 1997/1998 El Niño event and none during the 2015/2016 El Niño event but again flooding during its anomalous prolongation that occurred during 2017 [57]. Instead, the upper Moquegua

Valley (southern Peru, Figure 1) experienced torrential rain and flood during the 1982/1983 El Niño event, but no significant surface process occurred 15 years later [7,22,47].

In arid regions, a landscape response to rainfall events is typical. In fact, severe precipitations "*are more likely to cause landform change than are floods of similar magnitude elsewhere*" [58]. In contrast, weathering and erosion rates are orders of magnitude lower than in less dry environments, and soil development processes differ markedly from those found on the vast majority of the Earth's surface [5,59,60]. As a result, in coastal Peru, landscape modifications are slow compared to non-desert regions, except during major alluvial and eolian events [61–63]. With a multi-century recurrence interval for precipitation events, exceptional debris flows are reported in dry landscapes [64,65]. Under favorable conditions, layers deposited as a result of high magnitude flood events can be used as horizon markers in ancient landscape reconstruction. Slack-water deposits are significant examples of such useful beds [66]. In the last century, an event that produced a notable landscape impact in the valleys of northern Peru was El Niño of 1925/1926. For what concerns Moche Valley (see Figure 1 for location), the river spilled over vast tracts of alluvial plain, destroying bridges and irrigation systems and threatening the stability of the *Huaca del Sol*, the great archaeological monument of the Early Intermediate Period (ca. 100–800 CE) [14].

## 3. Materials and Methods

The screening of the documents relevant to this review was conducted according to the PRISMA statement [30,31], which assists reviewers and meta-analysts in transparently reporting why the review was conducted, what the authors accomplished, and what they discovered. To identify and select the documents, Scopus and WoS databases were processed [67–69]. To increase the chances to find useful documents, the topics "Ecuador" and "Chile" were added in the search modes. Moreover, the tag "*" was used in the search phrases to cover as many keyword combination as possible (Table 1). The databases were last consulted on 30 August 2022.

**Table 1.** WoS/Scopus database search modes.

| | Search Field | Search Phrase | Document Type |
|---|---|---|---|
| WoS | TS = Topic (Title, Abstract, Author Keywords, and Keyword Plus) | (((TS = ("El Nino*" OR Nino* OR ENSO OR "El Nino Southern Oscillation")) AND TS = (Peru OR Ecuador OR Chile)) AND TS = (landscape* OR settlement* OR site* OR archaeolog*)) | Articles, Review Articles, Proceedings Papers, Early Access, Book Chapters |
| Scopus | TITLE-ABS-KEY (Article Title, Abstract, and Keywords) | (TITLE-ABS-KEY (El Nino* OR Nino* OR ENSO OR El Nino Southern Oscillation) AND TITLE-ABS-KEY (Peru OR Ecuador OR Chile)) AND TITLE-ABS-KEY (landscape* OR settlement* OR site* OR archaeolog*)) | Article, Conference Paper, Conference Review, Review, Book Chapter |

Despite their great advantages, Scopus and WoS platforms may have introduced biases that favor Natural Sciences, Engineering, and Biomedical Research at the expense of Social Sciences, Arts, and Humanities. In a similar manner, documents written in English predominate over those written in other languages [68]. In any case, to find further articles and other works pertinent to the review, a careful examination was made of the reference list of the documents whose full text was examined. For the full-text analysis, basic criteria to establish how consistently literature data support the occurrence of El Niño events had to be defined. Since the review deals with geological and archaeological *data*, epistemological features of the respective disciplines (see [70–72] for geology and [73–75] for archaeology), as well as of the cross-disciplinary geoarchaeology [76,77], have driven the choice of these

criteria. The works cited above give particular attention to the conceptual meaning of the "data" in geology and archaeology and to the distinction between "data interpretation" and "data explanation" as a premise for the different perception of what counts as knowledge. Nevertheless, the peculiarity of the logical procedures inherent in the way of reasoning of geologists and archaeologists to interpret or explain the data is focused (see especially Frodeman [70] and Fogelin [74]). Accordingly, for the literature analysis, the following criteria were established and used: (a) geomorphological and stratigraphic features of the related deposits; (b) relationships between deposits and archaeological remains; (c) method of dating used to determine the age; (d) presence or absence of converging evidence; (e) consistency with data from other studies.

## 4. Screening Results

Among the documents extracted from the databases, only one duplicate was found, thus 458 works were identified in total (Table 2).

**Table 2.** Summary of identification, selection, and full-text examination stages.

|  | **WoS** |  | **Scopus** |
|---|---|---|---|
| Database mined documents | 383 |  | 306 |
| After duplicate check | 383 |  | 305 |
| Documents found in both databases |  | 230 |  |
| Identified documents |  | 458 |  |
| Excluded after abstract evaluation | 337 | 402 | 261 |
| Selected documents | 46 | 56 | 44 |
| From bibliography of selected works |  | 14 |  |
| Reviewed documents |  | 70 |  |

Then, careful abstract evaluation allowed the selection of 56 documents. A total of 402 documents were disregarded because they dealt with topics in other subject areas (i.e., medicine, public health, sustainability, ethnology, anthropology, oceanography, tectonic, geochemistry, psychic atmosphere, meteorology, glaciology, dendroclimatology, palynology, biology, zoology, botany, ecology, astronomy, history, economy, and sociology) or were irrelevant to the review's objective in space and/or in time (i.e., study cases not on the central Pacific coast of South America or outside the considered time). Fourteen documents identified by examination of the bibliography of the selected works were added for the review analysis (Table 2); thus, a total of 70 documents were finally reviewed (Tables 3 and 4).

**Table 3.** Summary of reviewed documents. BSD = Bibliography of selected documents.

| **Author(s) and Year** | **Source** | **Sites/Study Areas** |
|---|---|---|
| Nials et al., (1979 a, b) [14,15] | BSD | Huaca del Sol and 2 other sites (Moche Valley) |
| Samaniego et al. (1985) [78] | BSD | Cerro Sechin (Casma Valley) |
| Craig and Shimada (1986) [16] | Scopus | Batan Grande (La Leche Valley) |
| Sandweiss (1986) [79] | Scopus | Las Salinas (North coast of Santa mouth) |
| Rollins et al. (1986) [80] | Scopus | Las Salinas (North coast of Santa mouth) |
| DeVries (1987) [81] | BSD | *Review article* |
| Wells (1987) [82] | BSD | 7 sites (Casma Valley) |
| Wells (1990) [17] | BSD | 7 sites (Casma Valley) |
| Grodzicki (1992) [20] | BSD | Pampa de Nazca (Nazca Valley) |
| Moseley and Richardson (1992) [83] | BSD | Huaca del Sol (Moche Valley) |
| Moseley et al. (1992) [84] | BSD | Las Salinas (North coast of Santa mouth) |
| Ortlieb and Machare (1993) [85] | WoS, Scopus | *Review article* |
| Uceda and Canziani Amico (1993) [86] | BSD | Huaca de la Luna (Moche Valley) |
| Grodzicki (1994) [21] | BSD | Cahuachi and other 3 sites (Nazca Valley) |
| Keefer et al. (1998) [87] | BSD | Quebrada Tacahuay (Moquegua Valley) |
| Wells and Noller (1999) [88] | Scopus | *Review article* |

Table 4. Summary of reviewed works. This table continues from Table 3.

| Author(s) and Year | Source | Sites/Study Areas |
|---|---|---|
| Veit (2000) [89] | Scopus | *Review article* |
| Franco and Paredes (2000) [90] | BSD | Pachacamac (Lurin Valley) |
| Satterlee et al. (2000) [22] | BSD | Quebrada Miraflores (North coast of Moquegua mouth) |
| Magilligan and Goldstein (2001) [24] | WoS, Scopus | Rio Muerto (Moquegua Valley) |
| Sandweiss et al. (2001) [55] | WoS, Scopus | *Review article* |
| Van Buren (2001) [91] | WoS | *Review article* |
| Calderoni et al. (2002) [61] | Scopus | Mejia (Tambo Valley) |
| Huckleberry and Billman (2003) [92] | WoS, Scopus | Quebrada de los Chinos (Moche Valley) |
| Keefer et al. (2003) [93] | Wos, Scopus | *Review article* |
| Dillehay et al. (2004) [94] | WoS, Scopus | Los Mochicas del Norte Valleys |
| Federici and Rodolfi (2004) [95] | WoS, Scopus | Ensenada de Atacames, *Ecuador* |
| Keefer and Moseley (2004) [96] | WoS, Scopus | lower Moquegua Valley |
| Brooks et al. (2005) [97] | WoS, Scopus | Santa Rita (Chao Valley) |
| deFrance and Keefer (2005) [98] | WoS, Scopus | Quebrada Tacahuay (South coast of Moquegua mouth) |
| Eitel et al. (2005) [4] | WoS, Scopus | Quebrada Palpa (Grande Valley) |
| Zaro and Alvarez (2005) [99] | WoS, Scopus | Moquegua Valley and North coast of Moquegua mouth |
| Fabre et al. (2006) [100] | WoS, Scopus | Mollendo (North coast of Tambo mouth) |
| Machtle et al. (2006) [101] | WoS, Scopus | upper and middle Grande Valley |
| Manners et al. (2007) [50] | WoS | middle Moquegua Valley |
| Andrus et al. (2008) [102] | WoS, Scopus | *Review article* |
| Magilligan et al. (2008) [51] | WoS | 3 sites (Moquegua Valley) |
| Beresford-Jones et al. (2009a) [103] | Wos, Scopus | Samaca (Ica Valley) |
| Beresford-Jones et al. (2009b) [104] | Scopus | Samaca (Ica Valley) |
| deFrance et al. (2009) [105] | WoS | North and south coast of Moquegua mouth |
| Eitel and Machtle (2009) [106] | WoS | upper and middle Grande Valley |
| Mettier et al. (2009) [1] | WoS, Scopus | middle Piura Valley |
| Reindel and Wagner (2009) [107] | WoS | upper Grande Valley |
| Abbuhl et al. (2010) [108] | WoS, Scopus | middle Piura Valley |
| Beresford-Jones (2011) [109] | Scopus | Samaca (Ica Valley) |
| Bernal et al. (2011) [110] | WoS | Pastaza Valley, *Ecuador-Peru area of Amazon Basin* |
| Goldstein and Magilligan (2011) [111] | WoS, Scopus | upper and middle Moquegua Valley |
| Gayo et al. (2012) [112] | WoS, Scopus | Pampa del Tamarugal, *Chile* |
| Huckleberry et al. (2012) [113] | WoS, Scopus | middle La Leche Valley |
| Sandweiss and Kelley (2012) [114] | WoS, Scopus | *Review article* |
| Sandweiss and Quilter (2012) [115] | WoS | *Review article* |
| Winsborough et al. (2012) [23] | WoS, Scopus | Urpi Kocha Lagoon (Rio Lurin) |
| Etayo-Cadavid et al. (2013) [116] | WoS, Scopus | North and south coast of Moche mouth |
| Hanzalova and Pavelka (2013) [117] | Scopus | Ciudad Perdida de Huayuri (upper Grande Valley) |
| Macthle and Eitel (2013) [118] | WoS, Scopus | upper Grance Valley and middle Nazca Valley |
| Kalicki et al. (2014) [119] | Scopus | Lomas de Lachay (South of middle Huaura Valley) |
| Nesbitt (2016) [120] | WoS | Caballo Muerto Archaeological Complex (middle Moche Valley) |
| Caramanica and Koons (2016) [121] | WoS, Scopus | Pampa de Mocan (Chicama Valley) |
| Pavelka et al. (2016) [122] | WoS | Cantalloc (upper Nazca Valley) |
| Christol et al. (2017) [123] | WoS, Scopus | West coast of La Leche mouth |
| Wang et al. (2017) [124] | WoS | Salar Grande, *Chile* |
| Kalicki et al. (2018) [125] | WoS, Scopus | Lomas de Lachay (South of middle Huaura Valley) |
| Delle Rose et al. (2019) [26] | WoS, Scopus | Cahuachi (middle Nazca Valley) |
| Caramanica et al. (2020) [126] | WoS, Scopus | Pampa de Mocan (Chicama Valley) |
| Kalicki and Kalicki (2020) [127] | WoS | Lomas de Lachay (South of middle Huaura Valley) |
| Sandweiss et al. (2020) [13] | WoS, Scopus | *Review article* |
| Uceda et al. (2021) [128] | WoS, Scopus | *Review article* |
| Sandweiss and Maasch (2022) [129] | WoS, Scopus | *Review article* |
| Rubinatto Serrano et al. (2022) [130] | WoS, Scopus | Rio Muerto and 3 other sites (Moquegua Valley) |

Thirteen review articles are present among the documents whose full text was analyzed (Tables 3 and 4). None of these focused on landscape modifications due to El Niño events along the whole coastal Peru. Moreover, no useful data for this review were found in twenty reviewed works, including three articles on Ecuador and Chile case studies

(Table A1 in Appendix A). The major subject in archaeology studies centers on human response. The reviewed archaeological studies mainly aimed to understand the occupation history of sites or the function of issues related to monumental buildings, where the recovery of deposits ascribable to El Niño was apparently incidental. However, useful insights, for the purpose of this review, can be argued also from such literature. To expose and discuss the findings, data on landscape changes extracted from the reviewed documents were grouped and related according to main study areas (Section 5).

## 5. Main Study Areas for Landscape Changes

### 5.1. Los Mochicas del Norte Valleys

This area includes the valleys between Rio La Leche and Rio Jequetepeque and has an ethno-historical significance [131]. The earliest work identified by citation database queries is the one of Craig and Shimada [16] on the Batan Grande archaeological complex (see Figure 1 for location). Recent Quaternary stratigraphy, analysed along a modern hydraulic excavation, suggests to the authors that few deposits survive from the 1925/1926 El Niño event, except for slack-water beds roughly dated between 650 and 1000 CE by associated funerary pottery. Such deposits accumulate in areas of reduced velocity during flood flows and may be related to episodes of fluvial morphological adjustment and reshaping channel morphology [66]. Unluckily, the contained archaeological finds do not allow more precise dating, and thus the use of the above deposit as a horizon marker is prevented. Such a question may be addressed by absolute dating of several samples.

South of Batan Grande, along the Jequetepeque Valley (Figure 1), Dillehay et al. [94] documented several sediment release signatures of slack-water deposits containing Late Moche and Chimu ceramics and $^{14}$C dated between 415 and 1420 CE. These authors also documented different debris flow deposits as well as erosional truncation of floors at several archaeological sites, all likely associated with El Niño events. The multidisciplinary character of this study and the numerous radiocarbon dating of organic material extracted from alluvial deposits make the findings reliable. A long-term landscape shaping due to severe ENSO-related floods characterized the northern coastal desert before the arrival of the Spanish conquistadors. The human responses to the destructive effects of El Niño events are evaluated as "*highly sophisticated*" by the authors. Large rebuilding activities on damaged hydraulic structures, inferred by stratigraphic analysis at different archaeological sites, seem in fact to have allowed communities to avoid socio-economic repercussions of the hydrogeomorphic calamities. Similar results are obtained by Huckleberry et al. [113] for the inter-valley canal system named Racamuri (upper La Leche Valley, Figure 1), a millennial construction that reached its maximum extension between 900 and 1470 CE. Despite ENSO-driven floods and droughts, such a hydraulic structure would continue to work for centuries, likely even after the beginning of the colonial occupation. As a whole, for the northernmost valleys of coastal Peru, the selected literature data do not suggest drastic landscape changes or dramatic human responses to strong hydrogeomorphic episodes. Only the ca. 775 CE episode of abandonment of the Pampa Grande site (45 km southeast of Batan Grande) may be associated with a strong El Niño with good confidence [129]. It must be highlighted that this date is located within the time interval of the event postulated in Ref. [16]. Thus, a detailed geoarchaeological study on this case is suggested.

### 5.2. Huacas de Moche (Lower Moche Valley)

The settlement of Huacas de Moche is an early capital of the Moche state. It is located in the lower Moche Valley and includes the well-known buildings named Huaca del Sol and Huaca de la Luna (Figure 2). In their pioneering works, Nials et al. (a) [14] report "*the discovery of an El Niño catastrophe of a magnitude far greater and more devastating than all other such natural disasters striking the coast since the conquistadors first arrived in 1532. Transpiring about 1100 A.D., this prehistoric Niño was of unprecedented magnitude, and the devastation it wrought taxes the imagination of geologists and archaeologists alike*". Since about 1000 CE, the Chimu culture had completely transformed the desert landscape, building a complicated

system of reclamation channels over an area of tens of square kilometers. According to the authors (who have surveyed hundreds of kilometers of ancient waterways), all the channels experienced massive erosion and dropped from use before the beginning of the successive sub-cultural phase (a few tens of years later). This allowed an approximate dating of the hydrogeomorphic disaster, which was consequently called the Chimu Flood. Later, much of the reclaimed area would revert to desert in short time. However, the lack of absolute dating makes this reconstruction questionable.

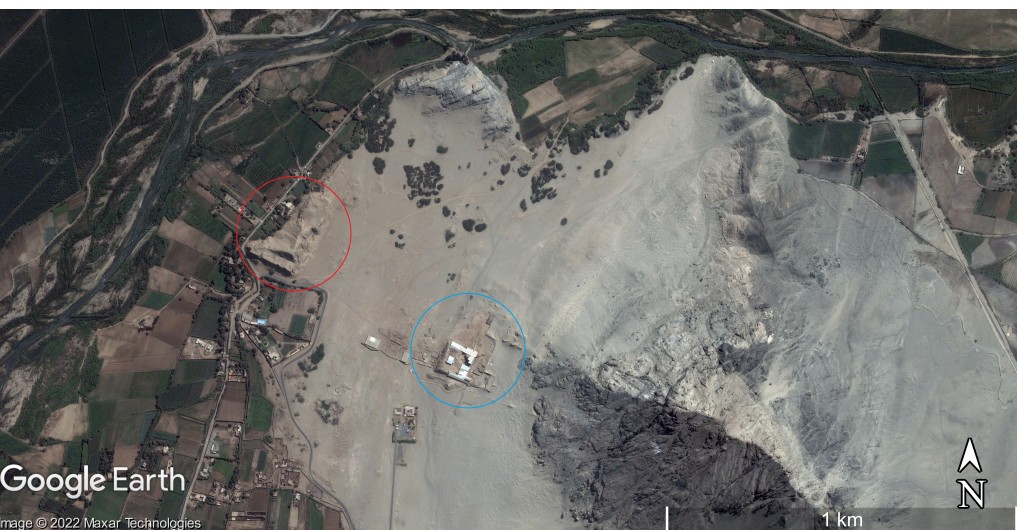

**Figure 2.** Huacas de Moche; 2004 satellite image (8°07′25″–8°08′22″ S, 79°00′17″–78°58′35″ W). Huaca del Sol and Huaca de la Luna are marked by a red circle and blue circle, respectively. The depositional origin of late Holocene layers covering the plain between the monumental buildings should be ascertained, taking into account the landscape processes argued by the authors [14,83,128].

The water level of the Chimu Flood would be still recognizable on the western side of the adobe brick pyramid *Huaca del Sol* (see Figure 7 on page 12 of Nials et al. (a) [14]), marked by a notch about 10–15 m above the present Rio Moche level. However, according to the geomorphological reconstruction of the authors, the major landscape change happened at the middle Moche Valley, 5–10 km upstream Huaca del Sol, with the lowering of the alluvial plain due to erosion estimated in ca. 5 m [15] (see Section 5.3). Moreover, the authors stated that "*a very conservative estimate would be flood waters at least 2 to 4 times the size of the 1925[/1926] floods, the worst in the last 400 years*" [15].

Several works selected from Scopus and WoS databases by the method described in Section 3 cite the works of the research group of Nials. Craig and Shimada [16] hint at a possible regional correlation with the slack-water deposit they found along the Rio La Leche Valley (see Section 5.1), thus laying the groundwork for the conceptualization of an 11th century El Niño event. Wells [17] suggests the possible coincidence of the Chimu Flood with the so-called Naymlap Flood, an ethno-historically recorded hydrological disaster associated with the name of a cultural hero [7,132]. Again, Wells and Noller [88] use the Chimu Flood to explicate the recurrence interval of the "Mega" El Niños in northern coastal Peru. Finally, Van Buren [91], Nesbitt [120], Huckleberry et al. [113], and Caramanica et al. [126] cite Nials et al. [14,15], simply to describe the destructive effects of the major El Niño events on the ancient irrigation systems and the consequences on the human communities. None of these studies deals with the reliability of the reconstruction provided by Nials et al. [14,15].

By archaeological excavations on *Huaca de la Luna*, Uceda and Canziani Amico [86] argued for the occurrence of moderate El Niño events before the Chimu Flood. They interpreted three layers of sediment with remnants of painted washed-off walls, found on successive floors of the temple, as evidence of intense precipitation and runoff processes, the most recent dated around 600 CE. This latter should have produced significant flow within

the braided streams of the Moche Valley without, however, causing either changes in the landscape or the abandonment of the settlement as asserted by Moseley and Richardson [83]. Instead, according to these authors, between 500 and 600 CE, "*flood water brought by El Niño struck the Moche capital, they leveled much of the city, stripping as much as 15 feet*" (about 4.6 m) "*off some areas. It is unclear if the magnitude of destruction reflects more than one El Niño events, perhaps exacerbated by an earthquake. The survivors rebuilt their city only to see it gradually inundated by sand dunes that swept inland after forming on the beach at the mouth of the Moche River*". It is apparent how the interpretation of geoarchaeological data coming from the same site can even lead to conflicting hypotheses [128]. Thus, the impact of the El Niño event around 600 CE on settlement and landscape at Huacas de Moche must be further investigated, especially with the aim of filling possible gaps in knowledge and addressing the above question.

The study by Prieto et al. [133] on a mass-sacrifice event discovered at a site of the lower Moche Valley, in addition to reporting a possible landscape process due to an El Niño event dated between 1400 and 1450 CE, is a remarkable example of the indissoluble connection between geological and archaeological data in the late pre-Hispanic coastal Peru. According to these authors, "*stratigraphic evidences suggests that the sacrifice was made following a heavy rain/flood that deposited a layer of mud on top of the clean sand in which the children and camelids were buried*". The conspicuous number of $^{14}$C dated samples ensures a high reliability of the chronological attribution.

### 5.3. Caballo Muerto and Quebrada de los Chinos (Middle Moche Valley)

Caballo Muerto Archaeological Complex and Quebrada de los Chinos are located in the middle valley of Rio Moche (Figure 1). The first includes the building named Huaca Cortada, from which data on the effects of El Niño events on settlement and landscape have recently been collected by Nesbitt [120]. According to this author, the whole archaeological complex "*is situated within an environment susceptible to El Niño flooding*". Due to its geomorphological setting, especially the alluvial fan located 10.5 km north-east of Huacas de Moche (Figure 3) may preserve significant geoarchaeological proxies.

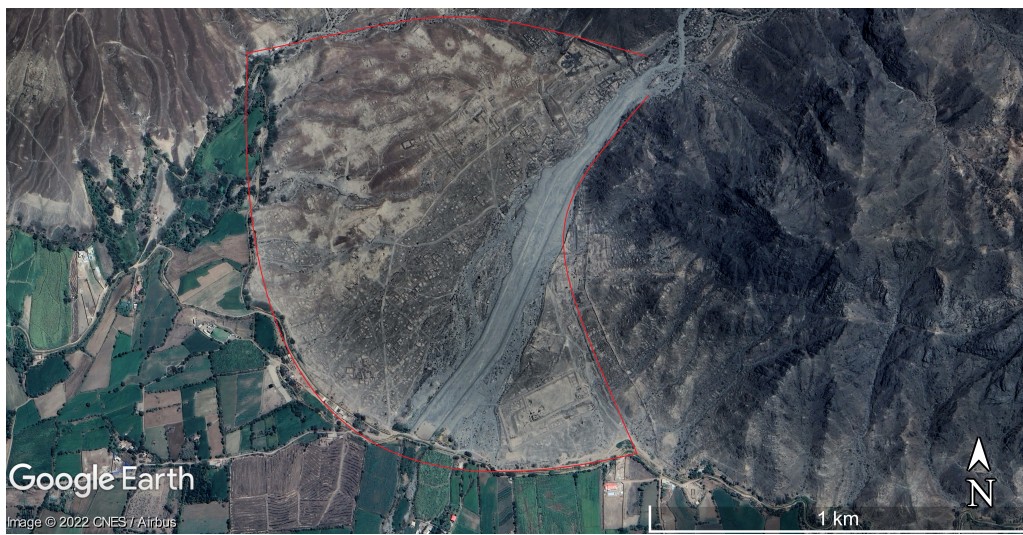

**Figure 3.** Caballo Muerto Archaeological Complex; 2021 satellite image (8°03′59″–8°04′53″ S, 78°55′19″–78°53′44″ W). The alluvial fan on which lies the main settlement of the complex is marked by a red line. Its stratigraphy should contain a wealth of data regarding El Niño events (see text).

Archaeological excavations at Huaca Cortada documented the occurrence of four El Niño events throughout the second half of the second millennium BCE. The identified El Niño proxies are laminated muddy layers deposited from runoff water. They contain thin laminas of white paint, which formed as the rain washed off the painted, plastered surfaces of the temple walls. These layers are sandwiched between pre- and post-event structures

and were exposed to weathering for a short time, as inferred from their sedimentological features. By the [14]C method, a date of 1600–1450 BCE is established for the earliest El Niño proxy of Huaca Cortada. The age of the subsequent events lies between 1100 and 900 BCE, as inferred from pottery finds. Finally, it must be noted that, as Caballo Muerto is surrounded by *quebradas*, changes in the shape of the braided streams for each event may be supposed, which would be significant modifications for a desert environment. Such a question should be addressed with geological studies. Nevertheless, according to Nesbitt [120], the communities of the Initial Period were able to rapidly reconstruct and enlarge buildings, and thus "*the social, religious and economic mechanisms that allowed for the mobilisation of labour [. . .] were not negatively impacted by El Niño*".

The geoarchaeological record of the middle Moche Valley preserves a frequent recurrence of El Niño also for the two millennia of the Common Era. Huckleberry and Billman [92] describe 13 ENSO-related flood and debris flow deposits beneath the present floodplain surface of Quebrada de los Chinos, which are younger than 2000 cal y BP. The findings of these authors seem to confirm the inference by Nials et al. [15] on apparent fluvial landscape changes due to severe precipitations (see Section 5.2), and reflect the late Holocene increases in El Niño activity [10,55]. With reference to methodological issues, the authors underline the need for a correct correlation between the geological characteristics of the deposits and the causative climatic events to make strong hypotheses [92].

*5.4. Casma and Cerro Sechin (Lower Casma Valley)*

With reference to radiocarbon dating, Wells [17,82] analysed in detail the alluvial terraces system, which characterizes the lower Rio Casma Valley (Figure 1). The system presents three floodplain surfaces whose ages of formation are between 3000 and 200 cal y BP (see Table 1 and Figure 4 of Reference [17], pp. 1135–1136), thus suggesting significant geomorphological changes during the late pre-Hispanic time driven by river sediment transport and tectonic uplift. These results are consistent with data on the beach ridge accretion of the Pampas Las Salinas [79,80], about 50 km northwest of the Rio Casma mouth. From a geological point of view, such a landscape dynamics is not surprising for valleys and coasts of the Peruvian desert. Late Pleistocene and Lower Holocene were, for example, earlier periods throughout which El Niños severely impacted coastal Peru, leaving signatures on ancient landscapes (see, e.g., References [19,85,93]).

Thirteen flood deposits younger than 3.2 ka are recorded along the stratigraphic section of Wells [17,82], six of which were deposited before the conquistadors first arrived in 1532. The upper two and the lower two of these latter are [14]C dated by means of incorporated organic material. The author, however, does not recognize the ca. 1100 event (i.e., The Chimu Flood) defined by Nials et al. [14] in the Moche Valley (see Section 5.2). She focus mainly on the recurrence of the hydrogeomorphic events rather than their strength and suggests, for the considered time interval, the increase in frequency of flooding and the occurrence of an event "*much larger than that which occurred during 1982–1983*" at least once every 1000 years. This hypothesis can be explained as follows: (a) the Mega El Niño events [19,80] actually exist and cause flood disasters in coastal Peru and possibly extreme climatic anomalies worldwide; or (b) the rainfall associated with the El Niño event is distributed such that extraordinary floods occur near Casma once every 1000 years [17]. Clearly, this question is crucial for the topic treated in the review.

Complementary evidence on the paleoflood record of the Casma district can be provided by archaeological data from Cerro Sechin site (Figure 1). However, only one laminated mud deposit ascribed to runoff water has been archaeologically dated (see Samaniego et al. [78]) and may be correlated to a terraced alluvial deposit [14]C dated at 1200 BCE by Wells [82]. Likely, new geoarchaeological research in this area would help in providing significant data on landscape change due to El Niño events.

### 5.5. Old Temple of Pachacamac and Urpi Kocha Lagoon (Lower Lurin Valley)

According to Franco and Paredes, the Old Temple of Pachacamac (Figure 4) was abandoned around 600 CE due to heavy rains that washed away and damaged the adobe walls of the building while runoff water deposited around thick layers of mud [90]. These authors argue that an unusual climate event triggered temple modifications and led to the development of new social trends and the introduction of architectural elements from the Andes Highlands. Later, Franco stated "*there is no doubt that the rains that caused this event correspond to a Mega Niño*" [134], which is now referred to also as the 6th-century El Niño event. The sedimentological and palynological study by Winsborough et al. on the near Urpi Kocha Lagoon (located 0.4 km west of Pachacamac) confirms such a suggestion [23]. These last authors found, in cores extracted from the bottom of the pool, evidence of three major floods associated with El Niño events in late pre-Hispanic time, the middle event dated between 436 and 651 CE (see Table 5 of Winsborough et al. [23], p. 611).

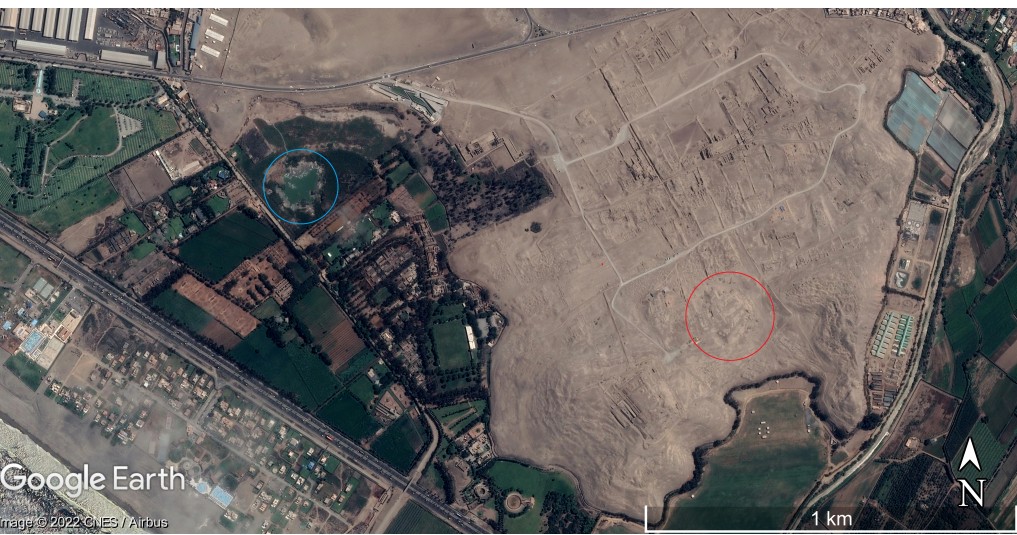

**Figure 4.** Pachacamac archaeological site; 2016 satellite image (12°15′08″–12°16′02″ S, 76°55′08″–76°53′31″ W). Old Temple and Urpi Kocha Lagoon are marked by a red circle and a blue circle, respectively. Archaeological and geological inferences on El Niño events were argued for by the authors [23,90,134].

The El Niño signature left at Pachacamac as recognized and interpreted by Franco and Paredes [90] is similar to the ones described by Samaniego et al. [78] and Wells [82] at Cerro Sechin (see Section 5.4) and by Nesbitt [120] at Caballo Muerto (see Section 5.3), respectively. It is apparent that, in late pre-Hispanic settlements, El Niño events may have left traces in the geoarchaeological record likely corresponding to landscape changes. Nevertheless, as carried out from further archaeological excavation, the adobe constructions of Pachacamac were seriously affected by repeated torrential rains up to the twentieth century, and especially by the 1925/1926 El Niño event [135]. The erosive processes caused by these rains have partially erased the traces of the most ancient events, making the reconstruction of the paleoflood record at the lower Lurin Valley more difficult.

The validity of the hypothesis that one, or even two, El Niño events affected the central coastal Peru around 600 CE is confirmed by the findings of Mauricio [136] at Huaca 20 site (Maranga archaeological complex, lower Rimac Valley, 30 km north-west of Pachacamac). Within such a site, now incorporated into the southern suburbs of Lima, the author describes two destructive floods separated by a phase of reconstruction during the Middle Horizon ([14]C dated between 550 and 690 CE). Despite the limited size of the site, several geoarchaeological sections showing the above sequence are reported as basic data.

Other areas in the lower Lurin Valley offer opportunities for future research on the relationships between landscape modifications and El Niño events in the Initial Period

and Early Horizon. Archaeological studies at the sites of Mina Perdida (5 km north-east of Pachacamac) and Manchay Bajo (10 km north-east of Pachacamac) have revealed the existence of debris flow, flood, and slack-water deposits in the respective geoarchaeological record [137,138]. A number of [14]C dating on organic remains extracted from the stratigraphic successions provide chronological references for the El Niño signatures.

### 5.6. Pampa de Palpa (Valleys of Rio Ica and Rio Grande)

The *pampa* southwest of Palpa (Figure 1) constitutes the northern margin of the Atacama desert, and was inhabited by several cultures during the pre-Hispanic times. It has been the object of different studies on the development of loess deposits, paleosoils, and alluvial terraces that provide insights about the long-term relationship between climate processes and desert landscape changes [4,101,106,118]. Moreover, this hyper-arid region includes relevant sites of interest for the present review and is next to the ceremonial center of Cahuachi (which will be discussed separately in Section 5.7).

Located at the lower valley of Rio Ica, the Samaca fluvial plain (Figure 1) is currently a riparian oasis that hosts numerous archaeological and paleobotanic remains. The latter are mainly constituted by partially fossilized trunks of the phreatophyte *Prosopis*, as established by Beresford-Jones et al. [103,104]. The geomorphological, archaeological, and paleoenvironmental evidence gathered by these authors allows them to state the occurrence of a major El Niño event "*at some time toward the end of the Early Intermediate Period, which spread a deep fluvial layer across the upper Samaca Basin, caused the river to cut some 5 m down into its floodplain and had catastrophic effects upon [a] canal system*" [103]. Such a result is consistent with the conclusions of other studies made in northern and central Peru [23,84,86]. However, it explicates part of the above landscape modification, as well as of the decline in the settlement. In fact, according to [103,104,109], human-induced gradual destruction of the *Prosopis* forest in pre-Hispanic time would have considerably increased the exposure to the flood hazard of landscape and settlement, making them more vulnerable to severe events. The deforestation process would culminate during the Middle Horizon, causing the final abandonment of the settlement [104].

Later, large urban settlements developed in the Late Intermediate Period (1000–1400 CE) within the *pampa* of Palpa, the most prominent example of what is the so-called *Ciudad Perdida de Huayuri* that, according to Hanzalova and Pavelka [117], was destroyed by an El Niño event. However, no indications or evidence of such an episode were found in other articles identified by citation database queries (Section 4, Tables 3 and 4). It must be observed that Eitel and their colleagues [4,106] posit an abandonment of such a "lost city" as a consequence of the depletion of water reserve, likely related to a climate change. The above questionable information in Ref. [117] may be due to a gap in the knowledge in the literature.

### 5.7. Cahuachi Ceremonial Center (Middle Nazca Valley)

According to Grodzicki [21,139], the archaeological site of Cahuachi (Figure 5) was affected by three catastrophic floods caused by El Niño between 2100 and 1000 BP. These studies support the major landscape changes ascribed to El Niño events in late pre-Hispanic time for the whole coastal desert of Peru. The second event would have even caused the collapse of the Nasca Culture (around 600 CE), while the third completely buried the ceremonial center [20,21]. In support of their hypothesis, the author describes conglomerate deposits that would result from deposition of exceptionally large, fluid debris flows. Finally, the landscape of Cahuachi would have been re-shaped (and the monumental building unearthed) throughout the last millennium. A necessary assumption for this argument is that abandonment and burial of the ceremonial center preceded the construction of the Nazca Lines [20]. However, such geoglyphs, created on the desert pavement by removing colored pebbles and leaving the underlying reg soil exposed, are mainly dated from 400 BCE to 600 CE [140–143].

Dealing with the second catastrophic flood inferred by Grodzicki [139], Silverman [6] observes that around 600 CE "*Cahuachi had ceased to function as the great early Nasca ceremonial center*". In their review on the occurrence of El Niño events, Machare and Ortlieb [56] consider the hydrogeomorphic events posited in References [20,21] no more than possible local climate change indicators. Nevertheless, the main question on the reliability of the Grodzicki's hypothesis comes from the research group headed by Eitel and Machtle [4,101,106]. Starting from the observation of "*the good state of preservation of geoglyphs, even where they cross valleys or erosion rills*", such authors "*disagree with ideas that catastrophic El Niño events destroyed the Nasca Civilization*" [4]. Again, as a result of extensive field surveys, they remark how the integrity "*of the line exemplarily illustrates that the valley floor has never been flooded since the Nasca period. Only a small channel in the foreground provides evidence for weak episodic runoff events during the past two millennia*" [106]. Finally, these authors also extend their conclusion to the contiguous *pampa* of Palpa (see Section 5.6).

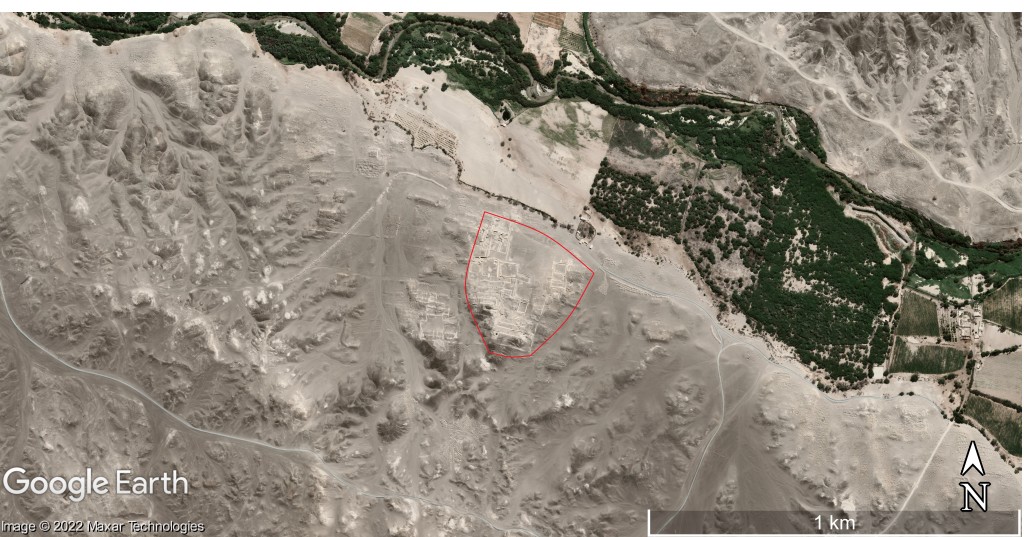

**Figure 5.** Cahuachi ceremonial center; 2022 satellite image (14°48′41″–14°49′30″ S, 75°07′45″–75°06′11″ W). The monumental building area investigated by Grodzicki [20,21], Orefici [144], and Delle Rose et al. [26] is marked by a red line.

To further test the consistency of the hypothesis of Grodzicki, stratigraphic and petrographic analyses of upper bedrock and surficial cover of the ceremonial center were carried out by Delle Rose et al. [26]. The results of this study show that the conglomerate deposits interpreted by Grodzicki as signatures of El Niño events, belonging instead to the Tertiary–Quaternary clastic succession that forms the regional substratum. Moreover, such coarse-size sediments are a source of pebbles and cobbles, which form the desert pavement. The inconsistency of Grodzicki's hypothesis was likely due to knowledge gap in geological stratigraphy [26].

Recent archaeological excavations on *Templo Sur* have shed new light on hydrogeomorphic events that damaged the adobe brick buildings at Cahuachi during the development of the Nasca Culture. In fact, the inferences of Orefici [144] about two torrential rains that partially destroyed roof and walls of the temple next to the end of the fourth century CE, a time with no El Niño events reported in the reviewed literature, lead us to reconsider both the climatic pattern responsible for the events and the type of Earth surface response expected for the coastal desert. It must be noted that samples gathered at Cantalloc (20 km east of Cahuachi) by Pavelka et al. [122], considered by these authors remains of an ancient flood, are [14]C dated between 47 and 480 CE.

### 5.8. Moquegua Valley and Quebrada Miraflores

Geoarchaeological studies throughout Moquegua Valley and surrounding areas [22,24,51,84] were mainly aimed to argue the relationships between El Niño events and human responses

rather than possible landscape processes [91]. Nevertheless, data exposed in the reviewed works allow one to argue for abrupt geomorphological changes to the desert environment. As a matter of fact, already thirty years ago, Moseley et al. [84] stated that "*serious landscape modification should indicate extreme ENSO conditions or ancient 'Mega-Niño' phenomena*".

Three main depositional units confidently ascribed to El Niño events have been identified and dated by the authors. Two of them have a late pre-Hispanic age, while the third (named Chuza Unit) is dated at 1607–1608 CE [96]. The oldest [14]C dated unit (730–690 CE) is formed by debris flow deposits (identified within the dry Rio Muerto sub-basin [24] (see Figure 6)), and flood deposits recognized elsewhere [51]. As it signaled fast and extensive regolith mobilization, this unit constitutes evidence of a singular hydrogeomorphic event. For what concerns the regolith production, in a highly seismic region such as coastal Peru, it is increased by the shattering of the landscape during the frequent earthquakes that produce pervasive ground cracking and microfracturing of hill-slope materials [84,96].

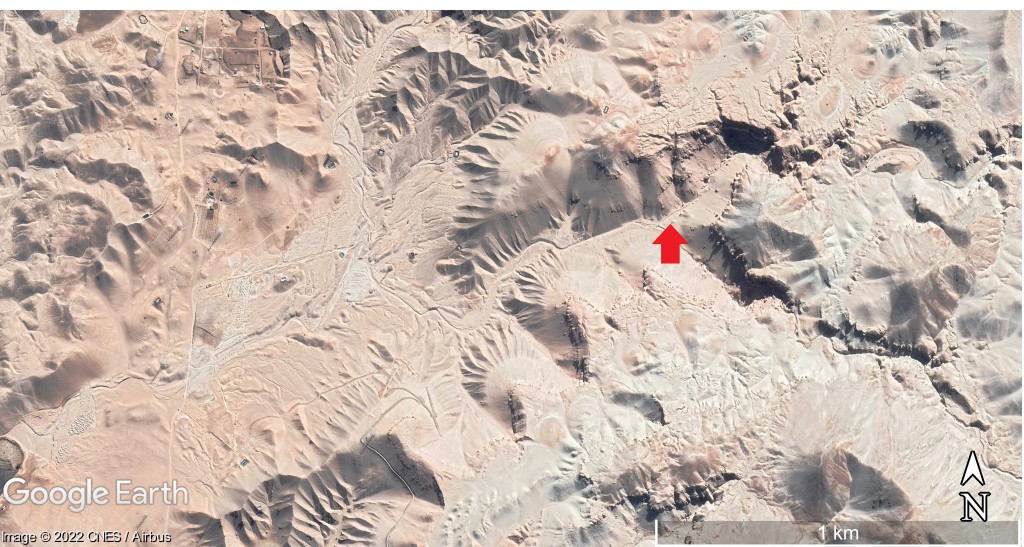

**Figure 6.** Rio Muerto sub-basin; 2020 satellite image ($17°17'58''$–$17°18'45''$ S, $70°59'11''$–$70°57'34''$ W). The sample point of the Miraflores Unit [51] is indicated by the red arrow (see text). Upstream, the cracked hill-slope area of regolith production may be observed.

The second depositional unit of interest for this review is a horizon marker referred to as the Miraflores Flood event [84]. It is formed by debris flow deposits and flood deposits indirectly dated between 1350 and 1370 CE by ice core data of the Quelccaya Ice Cap [22,145,146]. Such an age is confirmed by radiocarbon dates of samples taken at Rio Muerto [51] (Figure 6). At Quebrada Miraflores (Figure 7), the thickness of the marker reaches 1.2 m, which is a higher order of magnitude compared with any other flood deposits preserved in the geological record, including the Chuza Unit.

In this last area, mobilized sediments spread laterally out of the braided streams and up the ravine walls before descending to the sea across the coastal plain. Large clasts and boulders with a diameter of up to 3 m were also moved across the coastal plain by the Miraflores Flood [22]. Such a hydrogeomorphic event would also be implicated in the collapse or decline of the Chiribaya culture [111,147]. The imprint on the Peruvian desert surface of this 14th-century El Niño event is confirmed by zooarchaeological research. Rubinatto et al. posit "*that the abundance of anuran remains*" they found in Rio Muerto area may be related to the Miraflores Flood, since "*this event generated increased rainfall in the desert, creating conditions favorable for frogs and toads*" [130]. However, for a critical examination of such a hypothesis, these authors exhort further paleoenvironmental and zooarchaeological studies. In any case, the potential of Miraflores Unit as a horizon marker should be fully exploited in landscape reconstruction.

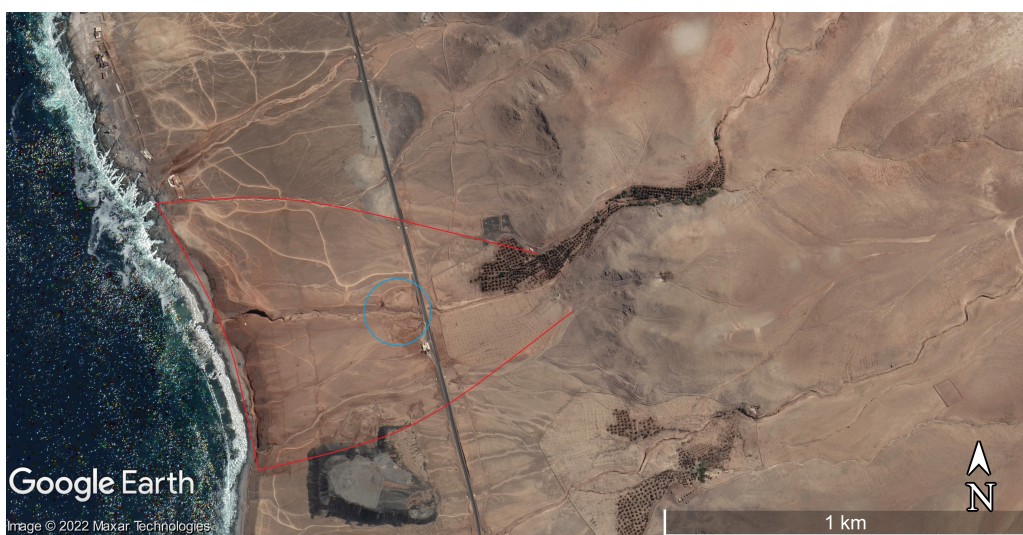

**Figure 7.** Quebrada Miraflores; 2021 satellite image (17°25′49″–17°26′36″ S, 71°22′22″–71°21′42″ W). The extent of the Miraflores Unit according to Satterlee et al. [22] is marked by a red line and the location of Chiribaya settlement by a blue circle (see text).

### 5.9. Discussion Summary

Landscape processes confidently ascribed to El Niño events are chronologically summarized in Table 5. They are considered responsible for changes in late pre-Hispanic coastal Peru to varying degrees.

**Table 5.** Chronology of late pre-Hispanic El Niño events argued as possibly responsible for landscape modifications.

| Date | Landscape Process | Basin or Site | Reference |
|---|---|---|---|
| ca. 1450 CE [1] | formation of alluvial terraces | Rio Casma | [17] |
| 1400–1450 CE [1] | mud deposition from runoff | Rio Moche | [133] |
| 1360–1350 CE [1] | debris flow activation, flood sedimentation | Q. Miraflores, Rio Moquegua, Rio Muerto | [22,24,51,84] |
| 1325 CE [1] | formation of alluvial terraces | Rio Casma | [17] |
| ca. 1100 CE [2] | flood sedimentation | Huacas de Moche | [14,15] |
| 1008–995 CE [1] | mud deposition from runoff | Pachacamac | [23] |
| 1000–650 CE [2] | flood sedimentation | Batan Grande | [16] |
| 730–690 CE [1] | debris flow activation, flood sedimentation | Rio Muerto, Rio Moquegua | [24,51] |
| 651–436 CE [1] | mud deposition from runoff | Urpi Kocka | [23] |
| ca. 600 CE [2] | mud deposition from runoff | Pachacamac | [18,90] |
| ca. 600 CE [2] | mud deposition from runoff | Huacas de Moche | [86] |
| 600–550 CE [2] | flood sedimentation | Samaca | [103,104,109] |
| ca. 0 CE [1] | formation of alluvial terraces | Rio Casma | [17] |
| 100 BCE [1] | debris flow activation | Rio Muerto | [51] |
| 900–1100 BCE [2] | mud deposition from runoff | Caballo Muerto | [120] |
| ca. 1200 BCE [1] | formation of alluvial terraces | Rio Casma | [17,82] |
| ca. 1450–1600 BCE [1] | mud deposition from runoff | Caballo Muerto | [120] |

[1] calibrated $^{14}$C age; [2] date inferred from archaeological remains.

The above chronology may be tentatively compared with paleo El Niño proxies. By analyzing a high resolution marine sediment record about 60 km west of Lima, Rein et al. [148] argued maximum of El Niño activity during the third and second millennium BP, while El Niño events would have been persistently weak during the Medieval Climate Anomaly (MCA, ca. 900–1300 CE). Yan et al. [149], computing SOI (Section 2) from different precipitation proxy records in Pacific Ocean, found negative index during MCA, which indicates general El Niño-dominated conditions. However, such differences in climatic reconstructions may reflect the geographic complexities of the Pacific coastal regions [45]. As a matter

of fact, SOI calculated for the Galapagos Islands (850 km off the coast of Ecuador) shows positive values during MCA in contrast with the ones of the western Pacific [149]. The oxygen isotopes ratio ($\delta^{18}$O) of ice core record from Quelccaya ice cap is depleted in $^{18}$O between 1100 and 1300 CE, thus highlighting a warming trend over the Andes [150]. Such a finding is consistent with low El Niño activity during MCA [148,151]. Consequently, the hydrogeomorphic events that can be attributed to El Niño with the lowest uncertainties should be those that precede or follow the MCA.

As a result of the WoS and Scopus data mining and the subsequent full-text analysis, the first evaluable hydrogeomorphic signature is located at Caballo Muerto and $^{14}$C-dated at about 3.5 ka BP [120]. However, it is in advance of the beginning of ENSO's modern periodicity (Section 2). Three more recent deposits dated around 3.0 ka and ascribed to El Niño events were also reported by this author (Table 5). To date, the magnitude of landscape response to these events along the middle Moche Valley is not yet investigated, while the only identified surface process is mud deposition from runoff over archaeological structures (Section 5.3). Moreover, the lower Lurin Valley also preserves geoarchaeological proxies [137,138] that can help shed light on the landscape modifications that occurred around the transition from the second to the first millennia BCE.

Around 1200 BCE, Casma Valley was affected by torrential rains, and thus, related proxies are preserved in both alluvial terrace sequences [82] and archaeological records of Cerro Sechin [78]. The hypothesis that both signatures were produced by the same El Niño event should be verified by further research (Section 5.4). According to Wells [17], the alluvial terrace system of the Casma Valley contains 13 flood deposits ascribable to late Holocene El Niños. Four $^{14}$C-dated events occurred in pre-Hispanic time, but unfortunately, no date fits well with events identified in other basins or sites (Table 5). In contrast, one or more events transpiring about 600 CE have hit the valleys of Rio Moche (Huacas del Sol), Rio Lurin (Pachacamac), and Rio Ica (Samaca) (Sections 5.2, 5.5, and 5.6). Such evidence has led some authors to speak of a 6th-century Mega El Niño (see, e.g., [21,23,134]).

Nevertheless, the recognition of one or more events to the transition from the first to the second millennium CE at some sites of north-central coastal Peru (Table 5) has strengthened the paradigm of the 11th-century Mega El Niño (see, e.g., [21,56,152]). The seminal reconstruction of the Chimu Flood by Nials et al. [14,15] has greatly influenced the literature, but it should be tested by further research, and the related deposits should be absolute-dated. Furthermore, the age of the ethno-historical Naymlap Flood remains unclear [84,132]. It ranges from around 1000 CE (see, e.g., [16,23]) to around 1350 CE (see, e.g., [17,153,154]). As a matter of fact, Naymlap Flood "*may refer to different floods in different valleys at different times during the Late Intermediate Period*" [7].

If on the one hand, El Niño events had catastrophic consequences associated with floodplain stripping, irrigation system damage, and settlement destruction, on the other hand, they also provided cultural opportunities, triggered technological innovation, strengthened community resilience against hydrogeomorphological events, and played important roles in replenishing and maintaining groundwater resources throughout the coastal desert [51,88,138]. Nevertheless, extensive human intervention (desert reclamation, *Prosopis* deforestation) has likely caused some variations in landscape response, also increasing exposure to extreme events of settlements and infrastructures (Sections 5.2 and 5.6).

Authors have deeply investigated the landscape response and destructive effects of the Miraflores Flood, the last hydrogeomorphic event ascribed to a "Mega" El Niño in pre-Hispanic times [22,24,51,84,111,147] (Section 5.8). With the arrival of Spanish conquistadors, the production of historical documents begins, so the sources of knowledge on El Niño events are enriched with a fundamental element [34,56,81,155]. However, it is only with the systematic observations of the warm ocean current running along the central Pacific coast of South America (El Niño in its initial meaning, see Section 2) that all the necessary criteria to assess El Niño events are available.

A final consideration must be made on the limitation of the above finding due to the use of WoS and Scopus. Papers published in journals not indexed in these citation databases

are obviously excluded. Moreover, as remarked upon in Section 3, documents written in English are favored over those written in other languages in data mining from WoS and Scopus [68]. Such problems may have partially affected this review. However, in the discussion on the main study areas (Section 5), documents written in Spanish and published in (not WoS- and/or Scopus-indexed) Peruvian journals and books ([131,134–136]) were also used, to the best of the author's knowledge. Clearly, the result may be hereafter improved and updated. On the other hand, data mining from other databases, systematic reading of papers written in Spanish and published in Peruvian journals, and the use of web search engines such as Google Scholar could be goals of future investigations.

## 6. Conclusions

Landscape modifications ascribed to El Niño events in late pre-Hispanic coastal Peru were mined and summarized by processing comprehensive databases (Section 3). Then, a review of the selected documents was performed in accordance with dependable guidelines (Section 4). The examination of the reliability of data contained in the reviewed documents allowed us to identify some critical questions in the treated topic and suggest several research goals (Section 5). The El Niño proxies discussed provide incomplete and heterogeneous paleoflood records for coastal Peru. The current state of geoarchaeological knowledge is not enough to define the temporal and spatial trend of the landscape effects of El Niño events during the late pre-Hispanic time. However, it gives some essential milestones (Section 5.9, Table 5).

Several landscape processes ascribable to El Niño events have been recognized by authors (Table 5). The less impressive one, mud deposition from runoff, has, however, sometimes been identified in archaeological sites, thus allowing valuable chronological attributions (Sections 5.3 and 5.5). Slack-water deposits found along the valleys of Rio La Leche and Rio Jequetepeque (Section 5.1), as well as other flood sediments recognized along the valleys of Rio Casma and Rio Samaca (Sections 5.4 and 5.6), and at the sites Huacas de Moche and Batan Grande (Sections 5.1 and 5.2), could have great potential in landscape reconstruction. The activation of debris flows has been recognized throughout Jequetepeque Valley (Section 5.1), Moche Valley (Section 5.3), Moquegua Valley, and the surrounding areas (Section 5.8). Within the catchment of Rio Muerto, regolith denudation and mobilization have shown a great size and extent (Section 5.8). The Miraflores Unit resulted in one of the major landscape changes for the whole coastal Peru, which was also involved in the end of the Chiribaya culture. It is apparently the better investigated proxy record confidently ascribed to an El Niño event (Section 5.8, Table 5).

Throughout the review, some possible research goals were highlighted that might be helpful in overcoming critical questions. They do not claim to be all-inclusive and are reported (not in the order of importance) in what follows: (a) to increase the number of the absolute dating of selected geoarchaeological records (see, e.g., the case of the deposits related to the Chimu Flood in the lower Moche Valley, Section 5.2); (b) to analyze stratigraphic successions that should preserve different proxies (see, e.g., the case of the alluvial fan within the Caballo Muerto Archaeological Complex, Section 5.3); (c) to investigate the landscape response of El Niño signatures discovered in archaeological sites (see the cases of Huaca Cortada and Cerro Sechin discussed in Sections 5.3 and 5.4, respectively); (d) to test the consistency of literature hypothesis with methods of various disciplines (see, e.g., the paleoecological and zooarchaeological study performed in the Rio Muerto catchment, Section 5.8); (e) to extract data from other databases also including documents written in Spanish and published in Peruvian journals (Section 5.9).

**Funding:** This research received no external funding.

**Acknowledgments:** I wish to thank G. Orefici, (the Centro de Estudios Arqueológicos Precolombinos (Peru)), for hospitality and valuable advice during research campaigns of 2012 and 2019 in Cahuachi.

**Conflicts of Interest:** The author declares no conflict of interest.

### Appendix A. Works Excluded from the Discussion by Full-Text Analysis

As explained in Section 4, 20 documents selected from database search were not discussed in Section 5 because they were not appropriate for the review's purpose (Table A1).

**Table A1.** Works whose full text was analyzed with no data useful for the objective of the review.

| Reference | Main Issue |
|---|---|
| [1] | Geomorphological effects of 1982/83 and 1997/98 El Niños |
| [61] | Morpho-pedological characterization of a Western Andean offshoot |
| [87] | Early-middle Holocene human-nature relationships |
| [95] | Late Quaternary evolution of a stretch of the Ecuador coast |
| [97] | Ancient peoples ability to mitigate El Niño effects |
| [98] | Debris flow burial episode of a Late Pleistocene site |
| [99] | Late pre-Hispanic desert reclamation and El Niño effects mitigation |
| [100] | Development of desert soil under ENSO conditions |
| [50] | Reliability of agricultural practices to face El Niño negative effects |
| [105] | Early-middle Holocene occupation history of a costal site |
| [107] | Introduction on methods and technologies in geoarchaeology |
| [108] | Geomorphological effects of El Niño on Western Andean Slope |
| [110] | Late Quaternary evolution of an Amazon Basin sector |
| [116] | Characterization of late Holocene coastal upwelling in Peru |
| [119,125] | El Niño influences on settlement pattern of a Western Andean offshoot |
| [121] | Paleobotanical analysis aimed to explore ancient desert reclamation |
| [123] | Reconstruction of the paleo-evolution of a coastal lagoon |
| [124] | Biological crusts effects on reg soil evolution |
| [127] | Late Quaternary environmental history of a Western Andean offshoot |

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
