# Peer review of "Landscape Modifications Ascribed to El Niño Events in Late Pre-Hispanic Coastal Peru"

_land, doi:10.3390/land11122207_

Round 1
Reviewer 1 Report
This article sets out to identify 'critical knowledge gaps' in El Niño event-impacted landscape studies; the method selected for this identification process is a broad, reference search to identify relevant articles and book chapters, and then a deeper analysis by the author of a selection of this research. I found this article a useful literature review of archaeological work around El Niño impact, but I do think there is a disconnect between the original aim set out by the author and the what the analysis actually provides.
The aim, I believe, is to highlight unanswered and critical questions about landscape transformation in areas of coastal Peru impacted by El Niño--I find this a very worthy and necessary exercise. However, the reader is presented with archaeological case studies, the vast majority of which were projects aimed at understanding the occupation history of sites, some, monumental huacas, where the recovery of El Niño-event related deposits was incidental. The result is the feeling that the literature review is somewhat tautological: by examining works whose research question was typically not landscape or ENSO-focused, are we not setting up a sample where those knowledge gaps are guaranteed? I suppose a more fundamental question is, why has the author turned his analysis to archaeological projects (as opposed to geological or geomorphological studies, climate science studies, sedimentary science, etc) and why, within the field of coastal Peruvian archaeology, the focus on site-based projects (as opposed to landscape-based, survey-style projects)?
I would hazard the guess that the method of collecting the state of the field via Scopus and WoS, has created a biased sampling of projects. Research done on monumental sites, or those with hypothetical 'collapse' scenarios, and those reports written in English, are more likely to be published in ranked journals. The author might consider expanding their net to include Spanish-language and Peruvian journals, informes or site-reports, and dissertations (Herbert Eling's thesis might be of particular use) to cover a more diverse set of case studies that might be more representative of landscape-based work (see also Vining et al. Dillon, Rundel on El Niño-blooming events, shifting ENSO regimes through the Middle Preceramic and Late Preceramic--Engel).
I'm also a bit skeptical (and this might be due to my own lack of expertise) of the author's identification of a diagnostic signature of Mega-El Niños in archaeological sites. Mud accumulation at sites built of un-baked adobe brick seems an imprecise proxy for the intensity of El Niño. Moreover, one of the studies cited by the author, the Billman and Huckleberry chapter, goes into great detail about the dangers of a weak correlation between intensity of the flood event and the thickness of flood deposit layers: according the Billman and Huckleberry, the depth of a flood deposit is dependent on the shape and altitude of the catchment basin, the location of the ravine in the valley (lower, middle, upper), and even the nature of loose gravels and other material in the ravine itself. Finally, Sandweiss et al.'s recent 2020 paper (also cited by the author) discusses the variety of 'flavors' of El Niño; identifying an impact signature is one thing, but ascribing it the descriptor 'Mega' is another.
There are important points made in this paper: for example, the occasional lack of nuance among archaeologists in describing geological data is problematic and worthy of attention, issues of dating, and of extrapolation.
Some specific line-items:
30-31: unclear what is being referred to by ‘due to the above’
110-111: Lower Chicama did experience flooding during the 2016-2017 Coastal El Niño event—although, those impacts have not been widely published in academic journals, see Rodríguez-Morata et al. (2019) The anomalous 2017 coastal El Niño event in Peru. Climate Dynamics 52:5605-5622; reports related specifically to Lower Chicama can be found in local news sources.
117-118: Does the author mean to refer strictly to alluvial transformation? Aeolian processes are both rapid and transformative in this area.
Table 1: Did the author find any discrepancies in their reference database search when using ‘ñ’ versus simply ‘n’ in the word Niño?
169: Correct spelling: Jequetepeque
Table 3: Pampa de Mocan is located in the Chicama Valley
337: It would be more accurate to refer to the site as Huaca 20, within the expansive complex of Maranga.
Author Response
Thank you for the effort and time you have spent to reviewing the manuscript. The replies to your comments are set out below.
Comment 1 . This article sets out to identify 'critical knowledge gaps' in El Niño event-impacted landscape studies; the method selected for this identification process is a broad, reference search to identify relevant articles and book chapters, and then a deeper analysis by the author of a selection of this research. I found this article a useful literature review of archaeological work around El Niño impact, but I do think there is a disconnect between the original aim set out by the author and the what the analysis actually provides.
Reply 1. In writing the first version of the manuscript, I thought that the expression 'critical knowledge gaps' could describe, succinctly and effectively, the aim of my work. By your comments and the ones of another reviewer, I understand that it was a wrong choice because it has caused some misunderstandings. Accordingly, in the revised version I have deleted such misleading expression and modified the introduction to unambiguously explain (I hope) the aim of the manuscript and give a clear connection with the results of the analysis.
Comment 2. The aim, I believe, is to highlight unanswered and critical questions about landscape transformation in areas of coastal Peru impacted by El Niño--I find this a very worthy and necessary exercise. However, the reader is presented with archaeological case studies, the vast majority of which were projects aimed at understanding the occupation history of sites, some, monumental huacas, where the recovery of El Niño-event related deposits was incidental. The result is the feeling that the literature review is somewhat tautological: by examining works whose research question was typically not landscape or ENSO-focused, are we not setting up a sample where those knowledge gaps are guaranteed? I suppose a more fundamental question is, why has the author turned his analysis to archaeological projects (as opposed to geological or geomorphological studies, climate science studies, sedimentary science, etc) and why, within the field of coastal Peruvian archaeology, the focus on site-based projects (as opposed to landscape-based, survey-style projects)?
Reply 2. Despite of the confusing previous “Introduction” (section 1 of the first manuscript version), you have well understood the scope of my review. Highlighting unanswered and critical questions can help in the hard task to outline the trends in space and time of the landscape effects of El Niño events (the overarching aim of the review). For such a purpose, any evaluable geoarchaeological data may be significant. As archaeological and geological data are indissolubly connected in the studies on the environmental changes of the pre-Hispanic coastal Peru, it is inevitable that many archaeological studies will be extracted by data mining, even if the ones not landscape or ENSO-focused. As the for the case of the Moquegua Valley and Quebrada Miraflores where the studies were mainly aimed to argue the relationships between ENSO and human responses (Section 5.8), data exposed in the reviewed documents can allow to argue landscape modification likely ascribable to El Niño events. This is not surprising since, as highlighted by Butzer (2008) ([76] in References) and some others, in settlements and other architectural sites the formation and degradation “mimic natural sedimentation and erosion”, so geoarchaeological data are informative also for surrounding landscapes. Nevertheless, also if the recovery of El Niño-event related deposits should be incidental, the data may be significant of the review purpose (see e.g. Samaniego et al. (1985) for the lower Casma Valley and Franco and Paredes (2000) for Pachacamac). Even if the recovery of El Niño-event related deposits was quite incidental during an archaeological study, it could be of great importance to outline the trends in space and time of the landscape effects of El Niño events. I have mined information from the databases with a transdisciplinary perspective. As you can read, useful studies belonging to different sub-disciplines may be identify (see e.g. Rubinatto Serrano et al. [130] in Section 5.8).
Comment 3. I would hazard the guess that the method of collecting the state of the field via Scopus and WoS, has created a biased sampling of projects. Research done on monumental sites, or those with hypothetical 'collapse' scenarios, and those reports written in English, are more likely to be published in ranked journals. The author might consider expanding their net to include Spanish-language and Peruvian journals, informes or site-reports, and dissertations (Herbert Eling's thesis might be of particular use) to cover a more diverse set of case studies that might be more representative of landscape-based work (see also Vining et al. Dillon, Rundel on El Niño-blooming events, shifting ENSO regimes through the Middle Preceramic and Late Preceramic--Engel).
Reply 3. About the biases due to the use of Scopus and WoS, in the first version of the manuscript I have only given some insights (lines 138-141). By virtue of your comment, in the revised version I have exposed more in deep such an issue (see the last paragraph of the new Section 5.9). In brief, the result of the present review may be hereafter improved and updated. However, data mining from Scopus and WoS, considered the "Titans of bibliographic information in today's' academic World" (see Pranckute, 2021; [69] in References) is an obligatory first step for review works. Data mining from other databases as well as systematic research in papers written in Spanish and published in Peruvian journals and the use of web search engines such as Google Scholar, may be consider next research objectives (lines 556-566, second version). Thanks also for your literature suggestions.
Comment 4. I'm also a bit skeptical (and this might be due to my own lack of expertise) of the author's identification of a diagnostic signature of Mega-El Niños in archaeological sites. Mud accumulation at sites built of un-baked adobe brick seems an imprecise proxy for the intensity of El Niño. Moreover, one of the studies cited by the author, the Billman and Huckleberry chapter, goes into great detail about the dangers of a weak correlation between intensity of the flood event and the thickness of flood deposit layers: according the Billman and Huckleberry, the depth of a flood deposit is dependent on the shape and altitude of the catchment basin, the location of the ravine in the valley (lower, middle, upper), and even the nature of loose gravels and other material in the ravine itself. Finally, Sandweiss et al.'s recent 2020 paper (also cited by the author) discusses the variety of 'flavors' of El Niño; identifying an impact signature is one thing, but ascribing it the descriptor 'Mega' is another.
Reply 4. Again due to the misleading previous “Introduction” (section 1 of the first manuscript version) I have caused this misunderstanding. The identification of a diagnostic signature of Mega-El Niños in archaeological sites is not my purpose. I have simply extracted and discussed data and case studies in which the so-called Mega-El Niño signatures can be significant to outline the trends in space and time of the landscape effects of El Niño events (see e.g. Miraflores Flood event). As a matter of fact, my previous works in such an issue (numbered [26] and [63] in the review) provide data in contrast with the hypothesis of the occurrence of three Mega-El Niños on Cahuachi as asserted by Grodzicki (1990,1992,1994). For the other points, I have added the correct observation of Huckleberry and Billman (2003) you noticed at the end of the Section 5.3. Clearly, I have quite in agreement with the statement of Sandweiss et al. (2020) you reported. On the other hand, it unambiguously rises from some geological literature (see e.g. DeVries, 1987, listed as [81] in my manuscript). I would like also to point out the highlighting of cases for which strong El Niño “provided cultural opportunities, … and played important roles in replenishing and maintaining groundwater resources throughout the coastal desert [51,88,138]” (lines 542-545). Thus, I am far from deterministic identification of diagnostic signature of Mega-El Niños.
Comment 5. There are important points made in this paper: for example, the occasional lack of nuance among archaeologists in describing geological data is problematic and worthy of attention, issues of dating, and of extrapolation.
Reply 5. Thanks again. However, in the pioneering works of Nials et al. (1979a,b), [14] and [15] in the review, the close collaboration between geologists and archaeologists (especially in the field) had already been indicated as mandatory. The problem you marked actually is present also regarding geologists that deal with archaeological topics (from my personal experience).
Specific line-items: 30-31: unclear what is being referred to by ‘due to the above’; 110-111: Lower Chicama did experience flooding during the 2016-2017 Coastal El Niño event—although, those impacts have not been widely published in academic journals, see Rodríguez-Morata et al. (2019) The anomalous 2017 coastal El Niño event in Peru. Climate Dynamics 52:5605-5622; reports related specifically to Lower Chicama can be found in local news sources; 117-118: Does the author mean to refer strictly to alluvial transformation? Aeolian processes are both rapid and transformative in this area; Table 1: Did the author find any discrepancies in their reference database search when using ‘ñ’ versus simply ‘n’ in the word Niño?; 169: Correct spelling: Jequetepeque; Table 3: Pampa de Mocan is located in the Chicama Valley; 337: It would be more accurate to refer to the site as Huaca 20, within the expansive complex of Maranga.
Brief replies: 30-31 (first version of the manuscript): the sentence has been edited; 110-111 (first version of the manuscript): I have integrated in the new manuscript both the reference to Rodríguez-Morata et al. (2019) ([36] in References) and the occurrence of flood at the lower Chicama Valleys during the Rodríguez-Morata et al. (2019) “anomalous” prolongation of 2016 El Niño, citing a new found in a web site (lines 113-114 of the new manuscript, [57] in References); 117-118: the sentence was edited according to your comment (lines 122-123, edited version); Table 1: No substantial discrepancy was found using ‘n’, nor new papers are added after abstract screening; 169: this error has been corrected (new line 186); Table 3: also this error has been corrected; 337: the sentence was edited according to your comment, thanks (new lines 369-370).
The main changes to the text are highlighted in the attached pdf file.

Reviewer 2 Report
This manuscript provides a useful review of the geological and archaeological record of the El Niño phenomenon in coastal Peru and one that will be of wide scholarly interest among scholars interested in paleoclimate in western South America. I commend the author on the research and extensive bibliography, which is impressive. However, prior to publication, some revisions are necessary to the manuscript to improve its argumentation and clarity.
Major comments:
1. The overall thesis of the manuscript needs to be clarified. In the abstract, the author states “The results provide knowledge synthesis in order to identify critical gaps and suggest specific research goals”. In my view the paper never explicitly states what some of the critical gaps are, nor in what the author thinks are specific (presumably future) research goals. This could be included in the concluding section.
2. Another problem with the article is that it is quite difficult in places to understand. The paper itself needs to be edited for clarity.
3. There is a confusing, but important statement on pp. 4 in the last paragraph of section 3 “These criteria were fixed based on the epistemological features of geology and archaeology”. It is unclear what the author is trying to explain here, but they cite Hodder, Butzer and others. I think it would be more beneficial to consider the broad literature within archaeology (in the Andes and elsewhere) that examines 1) the problems archaeologists (and earth scientists) face in reconciling the climate data with causation, such as culture change or “collapse” (in the words of Quilter/Sandweiss, shifting from collation, to correlation, to causation), and 2) the substantial literature dealing with how people respond to ancient climate change (both Caramanica’s PNAS article and the companion piece by Nesbitt 2021 deal with [for instance] ideas about resilience). Given the heavy reliance in this paper on archaeology it is worth mentioning that a major subject in El Niño studies centers on human response.
4. While the author presents a significant bibliography, they do miss a few key sources that are relevant to their paper. With respect to post CE 1000 ENSO events, the paper by Prieto et al. 2019 (in PlosOne) on a massive Chimu child (and llama) sacrifice that was pretty clearly undertaken during the heavy rains of an El Niño (including laminated muds) around CE 1450 is crucial to cite. For earlier time periods there is the work of Sandweiss et al. 2009 titled “Environmental Change and Economic Development in Coastal Peru between 5800 and 3600 Years ago” (PNAS) which describes ENSO sediments similar to those reported by Nesbitt 2016; and Uceda and Canziani, but in even earlier contexts. Likewise, there is the work by Richard Burger on (radiocarbon dated) debris flows at a second millennium BCE site in the central coast (near Pachacamac). This article was published in a special issue of Fieldiana Biology in 1998 and remains an important source on Initial Period ENSO. These articles are also of some relevance when discussing “the first evaluable hydrogeomorphic signature” (pp. 15).
Minor Comments
1. I would change the title from “Late pre-Hispanic” to simply “pre-Hispanic” since many examples of from much earlier times (i.e. Caballo Muerto, Cerro Sechín).
2. Pp. 2—Author uses term lomas to refer “hill ridges” on the western side of the Andes. To some extent this is correct, but the term also refers to a type of fog vegetation (which sometimes also blooms during the El Niño events).
3. Pp 3. “Early Horizon Period” should simply be changed to Early Horizon (with dates- c. 800-200 BCE).
4. Pp. 4. Early Intermediate Period should have assigned dates (c. 100-800 CE)
5. Pp. 6 & 16 Jequetepeque is spelled incorrectly.
6. Pp. 7. Line 195—what does the author mean by “hydraulic opera”?
7. Pp. 9- Huacas de Moche is spelled wrong.
8. Pp. 15. Confidentially should be confidently
Author Response
Thank you for the effort and time you have spent to reviewing the manuscript. The replies to your comments are set out below.
#. This manuscript provides a useful review of the geological and archaeological record of the El Niño phenomenon in coastal Peru and one that will be of wide scholarly interest among scholars interested in paleoclimate in western South America. I commend the author on the research and extensive bibliography, which is impressive. However, prior to publication, some revisions are necessary to the manuscript to improve its argumentation and clarity.
Reply #. When I draft the manuscript, I believed the phrase "critical knowledge gaps" might summarily explain the goal of my research. I acknowledge that it was a wrong decision because it led to some misunderstandings. As a result, I removed this misleading phrase from the amended version and edited Abstract and Introduction to clearly state the overall thesis of the manuscript and improve its argumentation and clarity. Please note that in the new version of the Introduction, the meaning attributed to “late pre-Hispanic” (lines 32-34) is cleared (i.e. the phase in which El Niño shows its current or “modern” features (cf. e.g. Sandweiss et al. (2020), [13] in References). Its start roughly coincides with the beginning of the Early Horizon (lines 80-81). The choice to extend the analysis up to the arrival of the Spanish conquistadores (roughly corresponding with the start of the so called Little Ice Age), is obviously related to the fact that, since then, there are also written documents to evaluate the episodes of landscape modifications ascribed to El Niños.
Major
Comment 1. The overall thesis of the manuscript needs to be clarified. In the abstract, the author states “The results provide knowledge synthesis in order to identify critical gaps and suggest specific research goals”. In my view the paper never explicitly states what some of the critical gaps are, nor in what the author thinks are specific (presumably future) research goals. This could be included in the concluding section.
Reply 1. The Abstract has been edited and the misleading sentence removed. However, some examples of possible gap in knowledge are highlighted in the manuscript (see lines 271, 411, 445-446). With regard to the future “specific research goals”, they are (explicitly or implicitly) suggested in the Sub-sections of the Section 5 (see e.g. last paragraph of 5.1; the lines 270-272; the caption of Figure 3 “stratigraphy should contain a wealth of data” and related text; and so on until the last clue now stated in the lines 564-566. Obviously, I have not the presumption to indicate "major research projects" but only to provide some insights. However, accordingly with your comments, in the concluding section, some suggested goals are re-highlighted (lines 592-604).
Comment 2. Another problem with the article is that it is quite difficult in places to understand. The paper itself needs to be edited for clarity.
Reply 2. The manuscript has been edited with the support of my institution's English expert. Please, let me eventually know what sentences would remain to be rephrased.
Comment 3. There is a confusing, but important statement on pp. 4 in the last paragraph of section 3 “These criteria were fixed based on the epistemological features of geology and archaeology”. It is unclear what the author is trying to explain here, but they cite Hodder, Butzer and others. I think it would be more beneficial to consider the broad literature within archaeology (in the Andes and elsewhere) that examines 1) the problems archaeologists (and earth scientists) face in reconciling the climate data with causation, such as culture change or “collapse” (in the words of Quilter/Sandweiss, shifting from collation, to correlation, to causation), and 2) the substantial literature dealing with how people respond to ancient climate change (both Caramanica’s PNAS article and the companion piece by Nesbitt 2021 deal with [for instance] ideas about resilience).Given the heavy reliance in this paper on archaeology it is worth mentioning that a major subject in El Niño studies centers on human response.
Reply 3. With the aim of containing the number of pages of the review I have write a short methodological section. In the new version of the manuscript I try to give some guidelines (lines 146-160). I think it is not appropriate to occupy more space in order not to weigh down the text. The literature used for the methodology ([70-77]) was chosen taken into account that the review deals with the landscape modifications to climatic events rather than the human response. Because the selection of geological, archaeological and geoarchaeological data is crucial for this type of review, I believe that the reader have to be informed about the epistemological issue related with “the way of reasoning of geologists and archaeologists to interpret or explain the data” (cf. Frodeman [70], Fogelin [74], and other authors). However, I agree that a heavy reliance in the review is on archaeology, so this aspect is now mentioned in Section 4 (lines 176-181).
Comment 4. While the author presents a significant bibliography, they do miss a few key sources that are relevant to their paper. With respect to post CE 1000 ENSO events, the paper by Prieto et al.2019 (in PlosOne) on a massive Chimu child (and llama) sacrifice that was pretty clearly undertaken during the heavy rains of an El Niño (including laminated muds) around CE 1450 is crucial to cite. For earlier time periods there is the work of Sandweiss et al. 2009 titled “Environmental Change and Economic Development in Coastal Peru between 5800 and 3600Years ago” (PNAS) which describes ENSO sediments similar to those reported by Nesbitt 2016;and Uceda and Canziani, but in even earlier contexts. Likewise, there is the work by Richard Burger on (radiocarbon dated) debris flows at a second millennium BCE site in the central coast (near Pachacamac). This article was published in a special issue of Fieldiana Biology in 1998and remains an important source on Initial Period ENSO. These articles are also of some relevance when discussing “the first evaluable hydrogeomorphic signature” (pp. 15).
Reply 4. Thanks for this important report. The study of Prieto et al. (2019) ([133] in the new References list) is now discussed in Section 5.2 (lines 273-280). Given its importance, the case is also reported in Table 5 (page 15). As a matter of fact, assuming that the mass sacrifice was indeed motivated by a catastrophic rain event, there is a high probability that it caused landscape modifications, and even some recognizable hydrogeomorphic signatures, in the surrounding areas. This insight provides a starting point for further geological research. Instead, the literature on the earlier time periods is not evaluable since the review address the time of the “modern” ENSO (the last three millennia). I will take this into account if I extend the analysis in a next review. With regards the works of Richard L. Burger on sites of the lower Lurin Valley, two documents are now included in the revised version of the review (referenced as [137] and [138]). They are mentioned in Section 5.5 (lines 374-380) and in section 5.9 (Table 5 and lines 518-520, page 16).
Minor
- I would change the title from “Late pre-Hispanic” to simply “pre-Hispanic” since many examples of from much earlier times (i.e. Caballo Muerto, Cerro Sechín).
- Pp. 2—Author uses term lomas to refer “hill ridges” on the western side of the Andes. To some extent this is correct, but the term also refers to a type of fog vegetation (which sometimes also blooms during the El Niño events).
- Pp 3. “Early Horizon Period” should simply be changed to Early Horizon (with dates-c. 800-200 BCE).
- Pp. 4. Early Intermediate Period should have assigned dates (c. 100-800 CE)
- Pp. 6 & 16 Jequetepeque is spelled incorrectly.
- Pp. 7. Line 195—what does the author mean by “hydraulic opera”?
- Pp. 9- Huacas de Moche is spelled wrong.
- Pp. 15. Confidentially should be confidently
Brief replies.
- I hope to have remove this problem associated with the unclear definition of the considered time interval as by the original manuscript version (please, see reply #).
- Thanks for clarifying the meaning of the term lomas. I believe that is unambiguously used in the manuscript.
- The change has been made.
- The change has been made.
- The change has been made.
- I apologize for this oversight. Correction has been made (“hydraulic structure”, see line 212 of the edited version).
- The change has been made.
- The change has been made.
The main changes to the text are highlighted in the attached pdf file.

Reviewer 3 Report
I think that the main goal of the paper has been achieved by criticizing each case study on what is missing and how can be improved. This publication will allow future studies in the area using different or more developed methodologies in order to better understand the effect of El Nino on Peruvian coast.
Nice presentation of the methodology. Do you have any historical sources about the events or they only appeared when Spanish arrived?
Nice overview on studies on how El Nino is reported in different sites.
It will be helpful if you add some photos (even from the publications) of deposits you described. This will make easier to reader to visualize the description
The table f5 can be improved: add a column where you describe (in numbers 1-5?) the dynamic of the event? and how each event affected the site (abandonment, partially destroy etc) and a column of cultures (i.e. Nasca) or periods (i.e. medieval, prehistory etc) and climatic events (i.e. MCA).
Replace numbers under 10 with words: line 296 -> six; line 426-> two
Author Response
Thank you for the effort and time you have spent to reviewing the manuscript. The replies to your comments are set out below.
Comment 1. I think that the main goal of the paper has been achieved by criticizing each case study on what is missing and how can be improved. This publication will allow future studies in the area using different or more developed methodologies in order to better understand the effect of El Nino on Peruvian coast.
Reply 1. Thank you for the corrected understanding of the manuscript.
Comment 2. Nice presentation of the methodology. Do you have any historical sources about the events or they only appeared when Spanish arrived?
Reply 2. For historical sources about the events see especially the following references: (1) Quinn,W.H.; Neal, V.T.; Antunez De Mayolo, S.E. El Nino occurrences over the past four and a half centuries. J. Geophys. Res. 1987, 92, 14449–14461; (2) Ortlieb, L. The Documented Historical Record of El Niño Events in Peru: An Update of the Quinn Record (Sixteenth through Nineteenth Centuries). In H. Diaz & V. Markgraf (Eds.), El Niño and the Southern Oscillation: Multiscale Variability and Global and Regional Impacts, 2000; Cambridge: Cambridge University Press, pp. 207-296.
Comment 3. Nice overview on studies on how El Nino is reported in different sites.
Reply 3. Thank you for your approval.
Comment 4. It will be helpful if you add some photos (even from the publications) of deposits you described. This will make easier to reader to visualize the description.
Reply 4. This would lead to copyright issues with a lot of time to spend that I don't have available. Sorry.
Comment 5. The table 5 can be improved: add a column where you describe (in numbers 1-5?) the dynamic of the event? and how each event affected the site (abandonment, partially destroy etc) and a column of cultures (i.e. Nasca) or periods (i.e. medieval, prehistory etc) and climatic events (i.e. MCA).
Reply 5. Your advice is correct, but it would lead to problems of interpretation on the events dynamic and of certainty on dating which is premature to address for all the cases in table 5. This improvement can be addressed in a forthcoming study.
Comment 6. Replace numbers under 10 with words: line 296 -> six; line 426-> two.
Reply 6. Replacing have been made. Thanks.
The main changes to the text are highlighted in the attached pdf file.

Round 2
Reviewer 1 Report
The author has addressed my comments and concerns; I do see in one instance, the Pampa de Mocan is still listed as a part of the Jequetepeque Valley, though I believe this is a small oversight.
Author Response
I have corrected the error in Table 4, row 48. I apologize for such an oversight.